# Changes in Public Sentiment under the Background of Major Emergencies—Taking the Shanghai Epidemic as an Example

**DOI:** 10.3390/ijerph191912594

**Published:** 2022-10-02

**Authors:** Bowen Zhang, Jinping Lin, Man Luo, Changxian Zeng, Jiajia Feng, Meiqi Zhou, Fuying Deng

**Affiliations:** 1School of Earth Sciences, Yunnan University, Kunming 650000, China; 2Faculty of Science, Dalian University for Nationalities, Dalian 116000, China; 3School of Tourism and Geographical Sciences, Yunnan Normal University, Kunming 650000, China

**Keywords:** ERNIE pre-training model, emotions, Coupla entropy, COVID-19, Shanghai

## Abstract

The occurrence of major health events can have a significant impact on public mood and mental health. In this study, we selected Shanghai during the 2019 novel coronavirus pandemic as a case study and Weibo texts as the data source. The ERNIE pre-training model was used to classify the text data into five emotional categories: gratitude, confidence, sadness, anger, and no emotion. The changes in public sentiment and potential influencing factors were analyzed with the emotional sequence diagram method. We also examined the causal relationship between the epidemic and public sentiment, as well as positive and negative emotions. The study found: (1) public sentiment during the epidemic was primarily affected by public behavior, government behavior, and the severity of the epidemic. (2) From the perspective of time series changes, the changes in public emotions during the epidemic were divided into emotional fermentation, emotional climax, and emotional chaos periods. (3) There was a clear causal relationship between the epidemic and the changes in public emotions, and the impact on negative emotions was greater than that of positive emotions. Additionally, positive emotions had a certain inhibitory effect on negative emotions.

## 1. Introduction

Major emergencies refer to natural disasters, accident disasters, public health incidents, and social security incidents that occur suddenly, cause or may cause serious social harm, and require emergency response measures. At the beginning of 2020, a large-scale outbreak of the 2019 novel coronavirus pandemic (COVID-19) impacted China socially and economically. In particular, the complete lockdown in Wuhan in January 2020 caused most industries in the country to suspend production. As a result, production, economic development, and social order were seriously affected [1]. To curb the spread of the epidemic, China adopted a series of strict measures including investigation, management, prevention, and control. After more than two years of prevention and control, the epidemic was effectively controlled, yet with continued periodic and sporadic cases and local clusters in several parts of the country. Every outbreak affected the public mood, leading to increased mental health problems. Currently, with the Internet penetration rate at 61.2%, any social public opinion triggered by an event related to the epidemic spreads virally and exponentially [2]. This greatly increases the psychological burden of citizens, increases the probability that people will suffer from anxiety and depression, and has serious impacts on their physical and mental health. This study selected Weibo texts with the topic “Shanghai Epidemic (2022-3-10-4-10)” as the data source to reveal the emotional changes in the public under the epidemic. Public sentiment is the perceived sum of individual emotions but may differ from an individual’s emotion. It is a short-term reflection of the thoughts or behaviors of individuals with a certain consensus. At the same time, public sentiment is also a reflection of the overall background of a society. Changes in public sentiment since the outbreak of COVID-19 have become a popular topic in academic circles. Li et al. [3] surveyed more than 3000 respondents in China and found that people who experienced higher levels of COVID-19 risk had weaker positive emotions and stronger negative emotions. Chen et al. [4] administered the Burnout Clinical Subtype Questionnaire (BCSQ-36), the 10-item Connor-Davidson Resilience Scale (CD-RISC-10), the self-rated 16-item Quick Inventory of Depressive Symptomatology (QIDS- SR16), and the Self-rating Anxiety Scale (SAS) and found that during the COVID-19 epidemic, burnout among healthcare workers was influenced by anxiety and depression. Psychological resilience was a mediating factor. Chang and Li [5] used Weibo texts as a data source, applied sentiment value, social network, and other methods to explore the spatiotemporal differentiation characteristics of public anxiety during the epidemic, and systematically evaluated the impact of public anxiety during the crisis. Li et al. [6] conducted a qualitative analysis of consumer responses on Twitter to capture the emotional responses of consumers to announcements made by tourism organizations during COVID-19. Jiang et al. [7] analyzed the spatial and temporal characteristics of the emotional responses of crowds to the epidemic through Weibo data, constructed a special emotional dictionary for the epidemic, and proposed an analysis framework for the spatial and temporal characteristics of the multi-dimensional emotional responses of urban user groups in the epidemic scenario. Kim et al. [8] applied the Linguistic Inquiry and Word Count method to analyze numerous COVID-19-related tweets in English and found three stages of public sentiment changes during COVID-19. Feizollah et al. [9] applied the Latent Dirichlet Allocation model to analyze the comments made by users on the Internet to clarify the emotional attitudes of the public towards the “halal vaccine.” Through the research of domestic and foreign literature, it was found that: first, most of the sentiment analysis during the epidemic focused on negative emotions [10], or began with positive and negative emotions, and there is little research on multi-dimensional emotions. Second, the research data sources may be divided into two categories. One category originates from the questionnaire scale [11,12]. This type of data is more convenient to process, but it cannot fully reflect the emotional changes in the respondents; it is difficult to obtain samples and the number of samples is small. The other category originates from web texts [13,14,15]. This type of data has a sufficient sample size and can objectively and accurately reflect public emotional changes, but data processing is more complicated. Third, In terms of research methods, most scholars use “constructing sentiment dictionary” and sentiment value calculation methods to study public sentiment intensity [16,17]. These methods can quantify public emotional intensity and eliminate the interference of subjective emotions. However, it cannot measure the emotional color from the overall tone of the commenter, and its accuracy is unsatisfactory [18].

To fill the above research gaps, this study used Weibo text as the data source, and adopted a novel method to achieve multi-dimensional emotion classification (gratefulness, confidence, sadness, anger, and no emotion). In addition, we constructed a sequence diagram of public sentiment, explored the changing laws and potential influencing factors of public sentiment, and studied the causal relationship between the COVID-19 pandemic and sentiment, as well as the topic focus of different sentiment periods. The results provide scientific advice and a reference for decision-makers on how to effectively deal with public emotional crises and establish an emotional crisis intervention model in major social and public emergencies in the future. This may significantly contribute to maintaining the stability of society and public sentiment during public emergencies.

## 2. Deep Learning in the Emotion Domain

Sentiment analysis is mainly divided into two categories: emotion recognition of text and emotion recognition of pictures and audio. The methods for sentiment analysis of texts can be divided into three key categories: based on a sentiment dictionary, based on traditional machine learning, and based on deep learning. The method based on deep learning is the current mainstream sentiment analysis method. Through deep learning, the sentiment analysis of sentences is carried to abstract features, thereby reducing the need for manually extracting features and can simulate the relationship between words with local feature abstraction and memory functions. Deep learning is a new field in machine learning research first proposed by Hinton et al. [19]. It is used in data mining, machine learning, machine translation, natural language processing, and multimedia learning. As a part of natural language processing, sentiment analysis has also transitioned from traditional analysis methods into the field of deep learning [20,21]. Nag et al. [22] found that music evokes different emotions in people, and thus applied convolutional neural networks to classify music according to different emotions, which was the first combination of deep learning and nonlinear-based classification algorithms. K et al. [23] applied deep learning to reader emotion research, verified the effectiveness and interpretability of deep learning in this research field through the Bi-LSTM attention model, and found that reader emotion may be related to specific words and named entities. Serrano-Guerrero et al. [24] constructed a deep learning architecture of bi-directional gated recurrent units with multi-channel convolutional neural network layers to detect the emotions of patient reviewers and understand their attitudes towards hospitals. This compensated for the inability of hospital assessment systems to detect patient emotions and provided directions for future research. Zhang and Li [25] proposed a multi-feature fusion hybrid neural network teaching speech emotion recognition model to discover the rules and details of emotions of teachers during the teaching process. This achieved rapid recognition and classification of emotions in the process of realizing intelligent teaching. Liu and Wang [26] constructed a mental health stress detection model with an emotion analysis function based on deep learning to understand the psychological stress and adverse emotional reactions of students and to provide an experimental reference value for research in related fields.

At the end of 2018, Google had built one such model, named Bidirectional Encoder Representations from Transformers (BERT), that outperformed nearly all existing deep learning models in several Natural Language Processing (NLP) tasks [27,28]. With its own advantages, researchers began to apply BERT to various research fields, including emotion research. Yong et al. [29] used BERT as a foundation and combined it with convolutional neural and bidirectional long short-term memory networks to mine potential customer sentiment in food reviews. They verified the advantages and accuracy of the fusion model. This contributed to textual sentiment mining research. Tanana et al. [30] applied BERT to explore the emotions generated during the psychotherapy process and its impact on the future, which can provide therapists with a better understanding of the psychotherapy process. Cui et al. [31] constructed a deep learning spatial emotion perception evaluation method based on social media check-in data through BERT and found that individual emotions are highly correlated with the type of activity space. This has important implications for decision makers in urban public safety, public health, and design management.

Enhanced Language Representation with Informative Entities (ERNIE) is a deep learning method for constructing language expressions proposed by the Chinese company Baidu based on BERT. It has shown better performance than other models on Chinese language processing tasks [32]. Xu and Hu [33] applied ERNIE to study the Weibo comments of tourists in the Qinghai-Tibet Plateau and found that the cognition of local residents was focused on emotional expression. Zhang et al. [34] combined the BiLSTM + attention + CRF model and ERNIE to study the public sentiment classification under the COVID-19 pandemic, solved the two tasks of sentiment dictionary expansion and sentiment classification, and further verified that ERNIE in Chinese text sentiment outperforms other models in classification.

## 3. Methodology

### 3.1. Study Design

This study was approved by the Human Research Materials Ethics Committee of Yunnan University (batch number: CHSRE2022020). The study applied deep learning methods to quantitatively analyze online reviews. It aimed to explore the changes in public sentiment during major public emergencies. Therefore, Shanghai, where the outbreak of the COVID-19 pandemic broke out suddenly, was chosen as the location for this study. Online comments were taken from the Weibo social networking platform. Additionally, we searched with the keyword “Shanghai epidemic” to obtain online comments related to the epidemic in Shanghai. Furthermore, by browsing the commenters’ IP addresses, we ensured that the online comments reflected the emotional changes in the locals in Shanghai. In the research process, we adopted manual screening to ensure the accuracy and applicability of the data. We also established a pre-training set in the form of manual labels to train the model and ensure the accuracy of the final classification results.

### 3.2. Data Sources

#### 3.2.1. Acquisition of Weibo Comment Data

Weibo is a popular social media platform in China similar to Twitter, and a Weibo comment is user’s opinion on a phenomenon. It can well reflect the real views and feelings of Weibo users regarding an event. We used “Shanghai Epidemic” as a keyword to conduct an advanced search on Weibo. The interval from 10 March to 10 April 2022 was also selected as a search parameter. Using Python to crawl the search content, 133,831 comments were obtained, including comment time, comment content, title connection, and commenter nickname, as shown in Table 1.

The acquisition of research data was approved by the Human Research Materials Ethics Committee of Yunnan University (batch number: CHSRE2022020) who supported the research.

#### 3.2.2. Data Preprocessing

To ensure that the Weibo data objectively, 
scientifically, accurately, and effectively reflected public emotions, the data 
were processed according to the following rules: (1) User location screening. 
Screening was performed on the IP addresses of Weibo comments and user 
profiles, and only the IP addresses displayed as Shanghai comments were 
retained. (2) Select private Weibo accounts. Private accounts allow individuals 
to effectively express personal emotions. (3) Eliminate data duplication. Weibo 
comments with duplicate content were removed. (4) Delete Weibo comments 
unrelated to the content of the epidemic. (5) Convert graphic expression 
symbols to text. Expression symbols can well reflect the actual emotions of the 
commenter [35]. For example, 
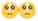
 represents 
sadness, 
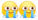
 represents bitterness, and 
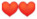
 represents gratitude. Accordingly, the expressions in the 
comments were converted into their text equivalent. (6) Punctuation and special 
symbols were removed to obtain higher accuracy for the ERNIE pre-training 
model. (7) We also checked the posting volume of Weibo accounts and the posting 
time of recent Weibo accounts to ensure that the users of Weibo and the Weibo 
accounts were not fake or fraudulent. After cleaning, 89,468 valid comment data 
were obtained. The response rate of the participants was 66.85%.

### 3.3. Methods

#### 3.3.1. Lexical Analysis of Chinese (LAC)

Lexical analysis is the most basic task of Chinese information processing, including automatic word segmentation, part-of-speech tagging, and named entity recognition [36,37]. This research used the LAC tool developed by Baidu Natural Language Processing Department to achieve automatic word segmentation of emotional texts. Lexical Analysis of Chinese (LAC), is a tool developed by the Baidu Natural Language Processing Department (NLP). It is a deep Bi-GRU-CRF network that can realize modeling word segmentation, part-of-speech tagging, and named entity recognition tasks [38], of which word segmentation is the most basic task, and the operation steps are mainly divided into the following three steps: [39,40]:

(1) Training corpus and annotations. This process requires researchers to manually label parts of speech in the pre-training set to train the word segmentation model.

(2) Character vectorization. According to the training process of the word segmentation model, before inputting the text sequence into the BI-GRU-CRF model for training, it is necessary to convert the text into a low-dimensional character vector to abstract the features of the word and use it as the input of the neural network for training. At the same time, the word vector obtained by pre-training can be used as input to achieve better word segmentation effect. To this end, it is first necessary to establish a Chinese character dictionary with a size of d × N, where d is the dimension of the word vector, N is the number of Chinese characters, and the word vector of each word was obtained by pre-training the method of word2vec. In this way, during word segmentation training, the input text can be queried against the dictionary to obtain the corresponding word vector, so that the text sequence becomes a real-valued matrix and is input to the neural network model for training.

(3) Dropout. Dropout is a commonly used method to prevent the overfitting of deep neural network training. Its basic principle is to temporarily discard network nodes from training according to a certain proportion, without updating their corresponding weights, but enable all nodes during the prediction process. In this study, a dropout layer was added after the two-layer GRU network to improve the performance of the word segmentation model.

The LAC tool used in this study is a relatively complete word segmentation tool. It achieved an accuracy of 95.5% on the test set and had a complete corpus data set. There was no need for manual labeling and pre-training, and only the required data was necessary and entered into (https://github.com/baidu/lac(accessed on May 1, 2022)).

#### 3.3.2. Keyword Weight Calculation

To characterize a specific type of emotion, based on emotion classification and lexical analysis, the term frequency–inverse document frequency (*tfidf*) weighting method was applied to weight the main keywords [41]. As a result of its strong universality and relatively simple and straightforward operation, the *tfidf* weight method has become one of the most important calculation methods in the text classification feature item weighting [42]. The specific calculation formula is as follows:(1)$${\mathrm{tf}}_{i,j}=\frac{n_{i,j}}{\sum n_{k,j}}$$
where ${\mathrm{tf}}_{i,j}$ is the word frequency, $n_{i,j}$ is the total number of times a certain word ($t_{i}$) appears in a certain type of emotional corpus data set ($d_{j}$), and $\sum n_{k,j}$ is the total number of words in the emotional corpus data set.
(2)$${\mathrm{idf}}_{i}=\mathrm{lg}\frac{\left|D\right|}{\left|\left\{j:t_{i}\in d_{i}\right\}\right|}$$
where ${\mathrm{idf}}_{i}$ is the file frequency, $\left|D\right|$ is the total number of microblogs in a certain emotional corpus data set, and $\left|\left\{j:t_{i}\in d_{i}\right\}\right|$ is the number of microblogs that contain the word $t_{i}$ in the emotional corpus data set.
(3)$${\mathrm{tfidf}}_{i,j}={\mathrm{tf}}_{i,j}*{\mathrm{idf}}_{i}=\frac{n_{i,j}}{\sum n_{k,j}}*\mathrm{lg}\frac{\left|D\right|}{\left|\left\{j:t_{i}\in d_{i}\right\}\right|}$$
where tfidf is the weight of word $t_{i}$ in corpus set $d_{j}$.

#### 3.3.3. ERNIE Pre-Training Model

Inspired by the BERT MASK strategy, ERNIE proposes a new language representation model. Compared with other models, ERNIE first models the prior semantic knowledge unit instead of directly learning the original language signal. This learning method can enhance the representation ability between model semantics, so that the model can learn the semantic representation of complete concepts. Additionally, the ERNIE model outperforms other models in five NLP tasks, including natural language reasoning, semantic similarity, named entity recognition, sentiment analysis, and question answering [32,43]. Therefore, this study chose the ERNIE model for fine-tuning to achieve high-precision classification of public sentiment.

#### 3.3.4. Estimating Transfer Entropy via Copula Entropy

Transfer entropy (TE) is a nonlinear method that uses the Markov property to detect information transfer and deduce the direction of causal relationships. The principle is to measure the probability that the dynamic change in process J affects the transition of process I through the transfer entropy I and J of two discrete and stationary processes. It is a quantitative measurement method. The key step is to estimate the Kullback-Leibler distance between transition probabilities [44]. However, TE calculations are complex and require sophisticated technical processing [45,46,47].

Copula Entropy (CE) is a new concept of entropy defined by Ma and Sun [48]. They proved that TE can be represented by CE. On this basis, a method for estimating TE by CE was proposed [49]. Therefore, estimating TE by CE can easily estimate the process and effectively obtain the temporal causal relationship between variables.

#### 3.3.5. Semantic Network Analysis

The semantic network analysis program Ucinet6.0 can analyzes the centrality, number of nodes, number of links, and deep network of each word, then visually analyzes the connection structure through NetDraw. In the visual semantic network analysis graph, keywords are represented in the form of circular nodes. The size of each circular node represents the centrality of the connection. The size of the node text represents the density of words and the links between nodes represent the strength of the connection [50]. In this study, Ucinet 6.0 was used to draw the semantic network diagram of the public in distinct emotional periods under the epidemic situation and calculate the centrality of high-frequency words of nodes to study the topics of public concern in these emotional periods.

## 4. Results

### 4.1. Data Analysis

First, to better grasp the content of the comments and conduct a reasonable sentiment classification, we used LAC to perform lexical analysis and drew a word cloud diagram (Figure 1). We selected the top 20 words in terms of word frequency and used tfidf for weighting to determine the word frequency and weighting of each segmented word. Results are shown in Table 2.

Table 2 shows that among the 20 most heavily weighted words, the words expressing the commenter sentiment were “Come on (tfidf=47.30),” “Sad (tfidf=28.26),” “Thanks (tfidf=22.54),” “Positive (tfidf=19.54),” and “Bitter (tfidf=17.18).” On this basis, this study combined previous research and text data characteristics of public emotions under the epidemic and divided them into five categories: gratitude, confidence, sadness, anger, and no emotion. The specific classification content is shown in Table 3.

Second, the ERNIE pre-trained model was applied for sentiment classification of text data. In this study, 10,000 Weibo texts were selected as the training set, with 2000 for each type of emotion. The training set was manually marked as a training sample for fine-tuning the ERNIE pre-training model. After experiments, this study set the maximum text length at 256 characters, the size of the training batch at 64, the learning rate at 5 × 10^−5^, and the number of training iterations at 11. Default values were selected for the rest of the parameters, which reached 88.6% on the test set. The training accuracy and classification of each emotion are shown in Table 4 and Table 5.

In Table 4, the accuracy rate represents the percentage of the correct prediction results in the total sample; the recall rate is the original sample, which means the probability that an actual positive sample will be predicted as a positive sample; and the F1 score considers both the accuracy rate and the recall rate. It is usually used to evaluate the comprehensive performance of a model. The higher the F1 score, the better the model effect.

Table 4 shows that the classification recall rates of gratitude, confidence, and sadness all exceeded 90%, the recall rate of anger was 84%, and the recall rate of no emotion was 75%. However, for the purpose of model testing, “no emotion” was used solely to improve the classification accuracy, so it would not interfere with the research owing to the low classification accuracy.

Table 5 shows that negative emotions accounted for the largest proportion (42.64%) of emotional comments, indicating that during the month after the outbreak of the Shanghai epidemic, negative emotions dominated public emotions, and anger (25.68%) dominated negative emotions. The no emotion category was the second largest (37.55%), indicating that nearly two-fifths of the public maintained an objective attitude. Positive emotions accounted for the smallest proportion (19.81%), only approximately one-fifth of the public maintained an optimistic attitude toward the epidemic. Most of the positivity was expressed as gratitude for the workers fighting the pandemic.

### 4.2. Temporal Changes in Emotion Classification

To determine the changes in public sentiment and potential influencing factors during the epidemic, we first drew the time series change chart of various emotions from 10 March to 10 April. Next, we queried the text data corresponding to various emotional extreme points and the daily number of new crown pneumonia cases. Finally, the four kinds of emotions (gratitude, confidence, sadness, and anger) were studied in stages. The timing charts of each emotion are shown in Figure 2, Figure 3, Figure 4 and Figure 5.

As shown in Figure 2, changes in public gratitude can be divided into three stages. First, from 10 March to 15 March, gratitude gradually increased. Although public gratitude fluctuated slightly, it generally showed a slow growth trend. The events that caused changes in gratitude at this stage included “residents helping each other” and “teachers caring for students,” and so on. Therefore, changes in gratitude at this stage were mainly affected by public behavior. The second state fluctuated repeatedly between 15 March and 1 April. Public gratitude remained largely unchanged overall but fluctuated frequently during this period. The events that caused changes in gratitude at this stage included “supporting Shanghai from many places” and “shops providing food for free,” and so on. Therefore, changes in gratitude at this stage were affected by public and government behaviors. The last phase was a rapid change from 1 April to 8 April. The public’s gratitude rose rapidly during the first half of the phase, then fell sharply by the end of the phase. At this stage, “medical staff from many places rushed to Shanghai” and “new cases continued to increase,” which caused the above changes in public gratitude. Therefore, changes in public sentiment at this stage were affected by government actions and the severity of the epidemic.

Figure 3 shows that changes in public sentiment can be divided into three stages. First, from 10 March to 19 March, events such as “determination of the transmission channel of the epidemic” and “government press conferences” caused a gradual upward trend in public confidence. Therefore, changes in public sentiment at that stage were affected by government actions and the severity of the epidemic. The second stage was from 19 March to 1 April. The continuous increase in “supporting Shanghai from many places” and “daily new cases” caused public sentiment to show a stage of rising volatility. Therefore, public confidence at this stage was affected by government actions and the severity of the epidemic. The last phase was the rapid change from 1 April to 8 April. Public confidence in the early stages of this phase grew rapidly. Changes were more erratic at the end of the phase. During this phase, daily new cases began to increase rapidly, leading to a decline in public confidence in fighting the outbreak. Therefore, public confidence at this stage was affected by the severity of the epidemic.

Figure 4 shows that public grief changes in two stages. The first stage was from 10 March to 29 March, with a gradual increase in sadness. Public grief fluctuated only slightly but maintained a slow growth trend as daily new cases increased. The negligence of the government’s initial epidemic prevention work caused people’s grief. Therefore, public grief at this stage was affected by the severity of the epidemic and the actions of the government. The second phase was the fast-rising phase from 29 March to 8 April. Public grief fluctuated several times, rising rapidly after each one. At this stage, daily new cases increased rapidly, and local government supplies began to run short. Residents feared for their lives. Therefore, public grief at this stage was affected by the severity of the epidemic and the actions of the government.

Based on Figure 5, the changes in public anger can be divided into two stages. The first phase was an uptick in volatility from 10 March to 1 April. This was in line with the increasing trend of daily new cases. In addition, incidents such as “doctors beating nurses” and “insufficient supplies” at this stage show that public anger was shaped by the severity of the outbreak and the behavior of the public and the government. The second stage was from April 1 to April 8. Incidents such as “destroying relief supplies,” “killing animals during the epidemic,” and “the number of new patients per day exceeded 20,000” caused public anger to rise rapidly. Thus, at this stage, public anger was shaped by the severity of the outbreak and public behavior.

In summary, it can be observed that changes in public sentiment during the epidemic were affected by government behavior, public behavior, and the severity of the epidemic. In the pre- and mid-epidemic period, public behavior and government behavior dominated public sentiment, while in the later stage of the epidemic, the severity of the epidemic gradually dominated public sentiment. Second, by analyzing the temporal changes of four specific types of emotions, we found that positive emotions were often divided into three stages in the process of change, while the change in and development of negative emotions was faster than that of positive emotions and only experienced two stages. It shows that negative emotions were more likely to be triggered than positive ones. Third, when analyzing Figure 2, Figure 3, Figure 4 and Figure 5, we found that around 8 April, when the number of new cases exceeded 20,000, the four types of public sentiment began to decline rapidly. Therefore, we speculate that when the existing state reaches a certain threshold, the public will gradually accept the status quo. To verify this speculation, we compared the time series changes in Weibo text with daily new cases (Figure 6) and found that the comment data about the “Shanghai epidemic” gradually declined after 8 April. This is consistent with our speculation. Fourth, our analysis shows that the changes in public sentiment under the epidemic can be divided into three periods: a period of emotional fermentation (slowly rising, fluctuating, and repeatedly fluctuating); a period of emotional climax (rapid change and rapid rise); and a period of emotional chaos after external conditions reached a certain threshold. In this study, that threshold appeared to have been reached when the number of daily new cases exceeded 20,000.

### 4.3. Analysis of Causality

Through the above analysis, we found that changes in public sentiment during the epidemic were related to changes in the epidemic. Whether positive emotions help to suppress negative emotions is also a question worth exploring [51,52,53]. As Diener [54] maintains, physical, sociocultural, individual experiences, environmental stimuli, and external environmental stimuli can all cause changes in personal emotions. We, therefore, proposed the first hypothesis:

**Hypothesis** **1.**
*The severity of the outbreak can cause changes in public sentiment.*


In addition, the revocation effect of emotion argues that [55] if positive emotions expand the individual’s thought-action sequence, they can also minimize the adverse effects of negative emotions on individuals; that is, positive emotions can “correct” and “undo” the after-effects of negative emotions. Therefore, we proposed the second hypothesis:

**Hypothesis** **2.***Positive public emotions will inhibit the generation of negative emotions during the epidemic*.

We represented the daily changes in epidemiological characteristics by the number of new patients every day. The daily changes in the public’s positive and negative sentiments were represented by the daily changes in the number of positive and negative sentiments (Table 6). When verifying the Hypothesis 1, we used the daily change in the epidemic situation as the dependent variable, and positive emotions and negative emotions as independent variables. When verifying Hypothesis 2, we used negative emotions as the dependent variable and positive emotions as independent variables and applied the CE method to estimate TE to verify the above hypothesis from the time series. If the result showed a transfer entropy value, it indicated a causal relationship, and the larger the transfer value, the stronger the causal relationship. Figure 7 and Figure 8 show the causal relationship between the epidemic and positive and negative sentiment. Figure 9 shows the causal relationship between positive and negative emotions.

From Figure 7 and Figure 8, it can be seen that there is an obvious transfer entropy value between positive and negative emotions and the epidemic, and there is a clear causal relationship between the epidemic and positive and negative emotions, confirming Hypothesis 1 [56,57]. Further analysis found that the impact on positive and negative sentiment peaked 27 days after the outbreak began. The difference is that the initial transfer entropy value of the epidemic to positive emotions was −0.346, and the maximum peak transfer entropy value was 0.402; while the initial transfer entropy value of the epidemic to negative emotions was −0.182, and the maximum peak transfer entropy value was 0.631. This shows that the impact of the epidemic on the negative emotions of the public was greater than the positive emotions in the early and later stages. This suggests that the public is more prone to negative emotions during major adverse health events.

From Figure 9, it can be seen that there is an obvious transfer entropy value between positive emotions and negative emotions, which indicates that there is a causal relationship between them, confirming Hypothesis 2. Furthermore, we found that the inhibitory effect of positive emotions on negative emotions fluctuated. The initial transfer entropy value was 0.306, and the maximum transfer entropy value (0.545) appeared 25 days after the epidemic. This signifies that within a month after the outbreak of the Shanghai epidemic, the inhibitory effect of positive emotions on negative emotions became increasingly stronger.

### 4.4. Social Network Analysis in Different Emotional Periods

To further explore public attention to the epidemic under different emotional periods, a keyword social network analysis was conducted by constructing a high-frequency keyword co-occurrence matrix [58]. In this study, the Ucinet 6.0 social network analysis software was used to calculate the centrality of network nodes and draw a social network relationship diagram [59].

#### 4.4.1. Emotional Fermentation Period

Network centrality is a common indicator to measure the influence of nodes in the network, indicating the importance of node status in the network. The relative centrality was arranged from high to low, and the top five keyword nodes were “epidemic (NrmDegree = 16.052),” “Shanghai (NrmDegree = 8.906),” “come on (NrmDegree = 8.837),” “sorrow (NrmDegree = 5.922)” and “Critical (NrmDegree = 5.614).” Among them, the relative centrality of “epidemic” was the highest. Combining the second-ranked keyword “Shanghai” and the fifth-ranked keyword “severe,” it can be observed that the severity of the epidemic in Shanghai was the focus of public attention during this period. In addition, the relative centrality of “come on” was much larger than that of “sad,” indicating that although the public’s positive and negative emotions coexisted during this period, positive emotions still dominated. In addition, the highly concentrated keywords such as "epidemic prevention", "local" and "China" in Figure 10 indicate that the positive emotions of the public come from the care and assistance of the government.

#### 4.4.2. Emotional Climax Period

The relative centrality was arranged from high to low, the first five keyword nodes were “epidemic (NrmDegree = 23.761),” “come on (NrmDegree = 9.716),” “help (NrmDegree = 8.899),” “people (NrmDegree = 8.899)” 8.415)” and “Material (NrmDegree = 8.256).” Among them, the relative centrality of “epidemic” was much larger than the centrality of other keywords and was greater than the relative centrality of “epidemic” during the emotional fermentation period, which signifies that the public’s attention to the severity of the epidemic in Shanghai increased during this period. Among the remaining four keywords, the relative centrality of “come on” and “help” was higher, which indicates that the public maintained a positive attitude in the face of the epidemic at that time. Furthermore, from the relatively high centrality of “materials” and the words “help” and “support” in Figure 11, it can be seen that the public’s positive emotions mainly derived from other regions’ support to Shanghai.

#### 4.4.3. Emotional Chaos Period

In descending order of relative centrality, the top 5 keyword nodes were “epidemic (NrmDegree = 15.809),” “nucleic acid (NrmDegree = 10.191),” “materials (NrmDegree = 10.074),” “Shanghai (NrmDegree = 9.988)” and “community (8.250).” Among them, although the relative centrality of the “epidemic” remained at the highest level, the relative centrality of the “epidemic” in both the emotional fermentation period and the emotional climax period declined, which shows that during this period, although the severity of the epidemic remained the focus of public concern, the level of attention started to decline. The remaining four keywords do not reflect the public sentiment, which indicates that it was difficult to change public sentiment due to the influence of the epidemic during this period. It can be seen from the words "nucleic acid", "material" and "community" in Figure 12 that during this period, the public paid more attention to their own daily security and safety.

In summary, we found that regardless of the emotional stage of the epidemic (fermentation, climax, or chaos), the change in the epidemic situation was always the core of public attention. With the change in emotional period, the positive emotions of the public began to disappear and were replaced by negative emotions. The public began to focus on their security and safety.

## 5. Discussion

### 5.1. Significance and Recommendations

Emotion is a complex psychophysiological process. Individuals have different physiological responses to different emotional stimuli. When people experience negative emotions during a crisis, this may be accompanied by deviant behaviors [60]. However, if a certain positive emotional experience is applied to an individual in time, the negative emotions will be suppressed, and their own behavior will also change [61], which is of great significance for ensuring the psychological and personal safety and social stability of the public during the epidemic. To ensure the generation of positive emotions and the reduction in negative emotions in public sentiment, we proposed the following suggestions:(1)The emotional fermentation period is generally at the early stage of the epidemic, where the impact of the epidemic is at its lowest and is not yet wide-ranging. At this time, prevention and control measures should be taken in a timely manner to ensure two-way communication and exchanges with the masses. The active publication of anti-epidemic events will help facilitate the public’s positive emotions to combat the epidemic and ensure the effective implementation of anti-epidemic policies. The public needs to lead by example, provide help and care to others, maintain a good attitude, and actively pay attention to the national government’s epidemic prevention policies and dynamic changes during the epidemic, but refrain from excessive remarks.(2)When the public mood reaches its climax, it signifies that the epidemic has begun to affect the normal lives of most local residents and has even begun to permeate to other provinces and cities. The focus should be on improving emergency medical treatment capabilities and material supply and demand matching capabilities, as well as the effective stabilization and standardization of markets. The media should publicize as much as possible, local anti-epidemic heroic deeds and the anti-epidemic assistance of other provinces and municipalities to reduce the focus on daily living issues and negative emotions caused by the epidemic. In addition, since most of the attention of the public is on the epidemic itself, the most obvious fundamental action to regulate the negative emotions of the public is to effectively control the spread of the epidemic.(3)When public sentiment reaches a chaotic stage, it signifies that the spread of the epidemic has exceeded most of the public’s expectations, and they gradually begin to reduce their attention on the epidemic and its corresponding events. The erroneous notion, “do not take measures and let the epidemic continue to develop,” gradually emerges in the hearts of the public. To guide citizens more effectively, it is necessary to strengthen epidemic prevention measures, reduce the number of new cases every day, and provide the public with actual data showing progress.

### 5.2. Limitations and Scope for Future Study

This study has some limitations. First, there are issues with data acquisition. Weibo is the largest social platform in China, with nearly 600 million users; however, most of the users are relatively young and middle-aged, while teens and older adult age groups are less represented. Therefore, the data represents only limited age groups. Second, the time span was short. This data was based on one month of content posted after the start of the pandemic in Shanghai, which leads to certain limitations in exploring the causal relationship between the epidemic and emotions, as well as the presence of positive and negative emotions. A third, limitation is whether a Weibo account can be identified as real by browsing the posted volume of Weibo accounts and the time of recent Weibo postings. Although most reviews reflect the real thoughts of users, sometimes, there may be hired writers in online reviews. Most of the posts they publish are not expressions of their own intentions but are influenced by the will of their employers.

To obtain more accurate research results in the future, network data could be combined with field research to include the emotional changes in people of all ages experiencing the epidemic. The time span of the data could be extended to include online data covering a 1-year period. The cause and effect of the epidemic and public sentiment could then be evaluated over a 1-month period. In this way, with the support of 12 months of data, more accurate and scientific research results can be obtained. Additionally, it would be advisable to obtain text data with location markers to analyze the public emotional distribution in space.

## 6. Conclusions

In this study, the ERNIE pre-training model was used to classify Shanghai public’s Weibo comments into five categories: gratitude, confidence, sadness, anger, and no emotion, to explore the temporal changes and potential influencing factors of different emotions. The study reached the following conclusions:During a public health emergency, public sentiment can be greatly affected. During the Shanghai outbreak, negative emotions dominated known emotional responses. In addition, among the negative emotions, sadness accounted for 16.96% and anger accounted for 25.68%. Therefore, anger was the primary negative emotion expressed;Public sentiment during the epidemic was affected by factors such as public behavior, government behavior, and the severity of the epidemic. In the pre-period, public behavior and government behavior dominated public sentiment. Later, the severity of the epidemic gradually dominated public sentiment;From the perspective of time series changes, the changes in public sentiment during the Shanghai epidemic can be divided into three periods: the emotional fermentation period, the emotional climax period, and the emotional chaos period. Through social network analysis, it was found that the epidemic has always been the core of public attention. However, as the emotional period changed, the positive sentiment of the public began to fade. Instead, the public became concerned about their own safety and security.The impact of the epidemic on the negative emotions of the public was greater than on the positive emotions, indicating that the public is more likely to experience negative emotions during major adverse health events. In addition, a causal relationship between positive emotions and negative emotions was detected, indicating that positive emotions have a certain inhibitory effect on negative emotions.

This study has implications for analyzing multi-dimensional emotional responses to major adverse health crises. It may also have practical implications for predicting the sequential stages of emotional responses to such crises and allow public officials to gain insight in how best to respond to major crises in a way that bolsters the public’s emotional wellbeing.

## Figures and Tables

**Figure 1 ijerph-19-12594-f001:**
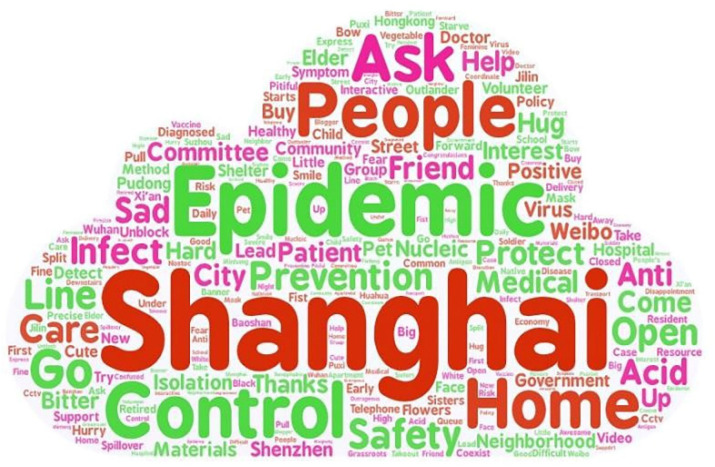
Word cloud map of the Shanghai epidemic.

**Figure 2 ijerph-19-12594-f002:**
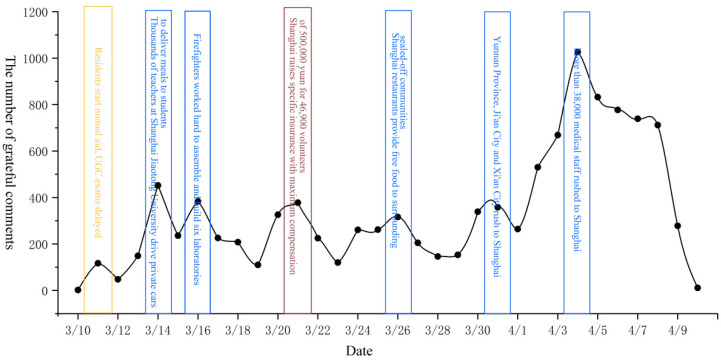
Time series change diagram of comments expressing gratitude.

**Figure 3 ijerph-19-12594-f003:**
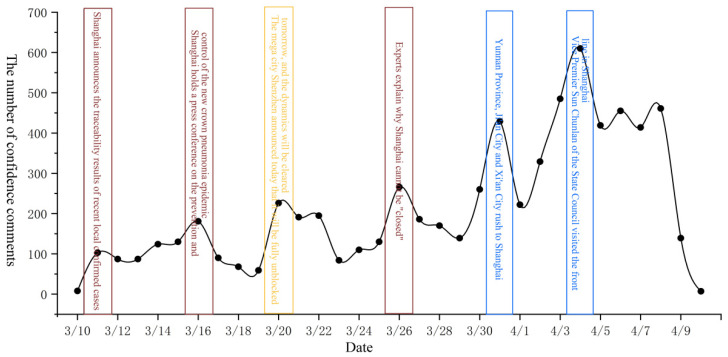
Time series change diagram of comments expressing confidence.

**Figure 4 ijerph-19-12594-f004:**
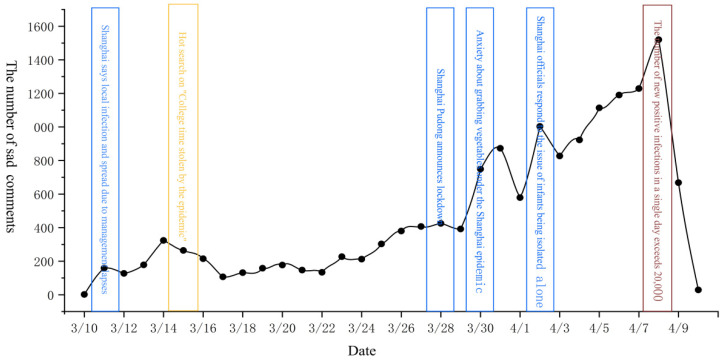
Time series change diagram of comments expressing sadness.

**Figure 5 ijerph-19-12594-f005:**
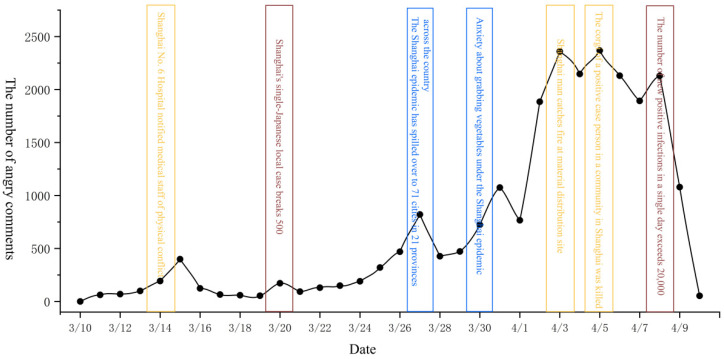
Time series change diagram of comments expressing anger.

**Figure 6 ijerph-19-12594-f006:**
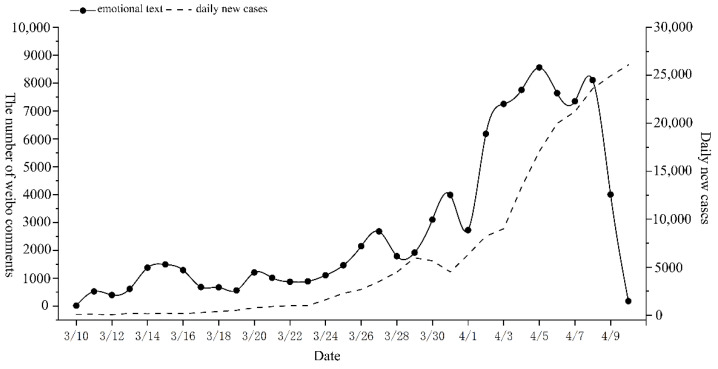
Number of Weibo comments and the time series change in daily new cases.

**Figure 7 ijerph-19-12594-f007:**
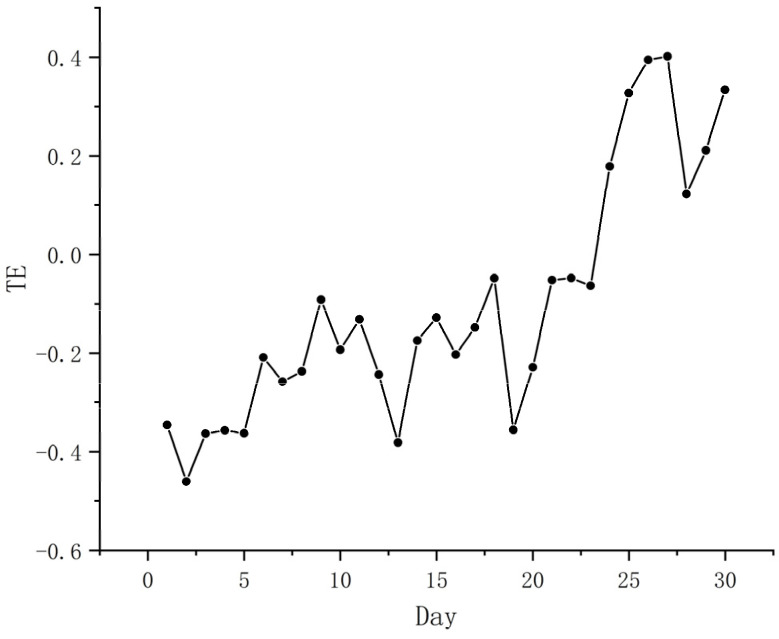
Causal relationship between the epidemic and positive emotions.

**Figure 8 ijerph-19-12594-f008:**
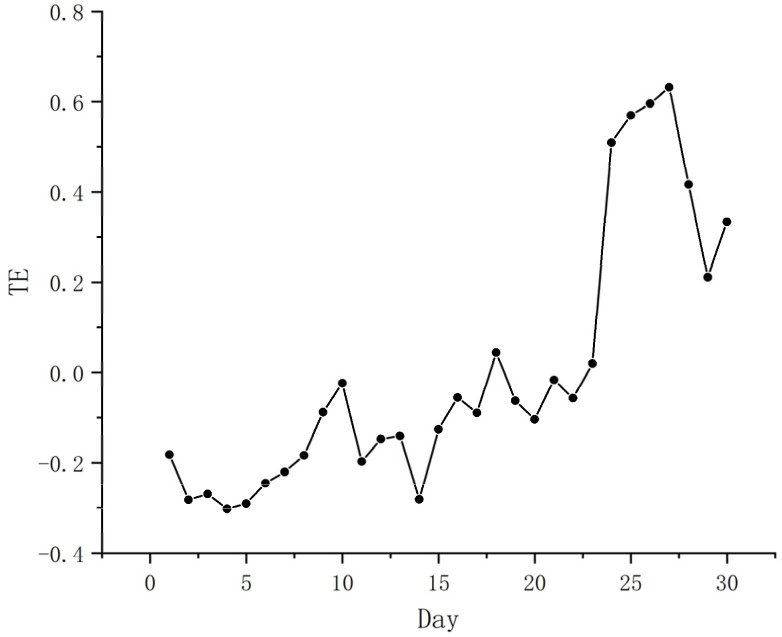
Causal relationship between the epidemic and negative emotions.

**Figure 9 ijerph-19-12594-f009:**
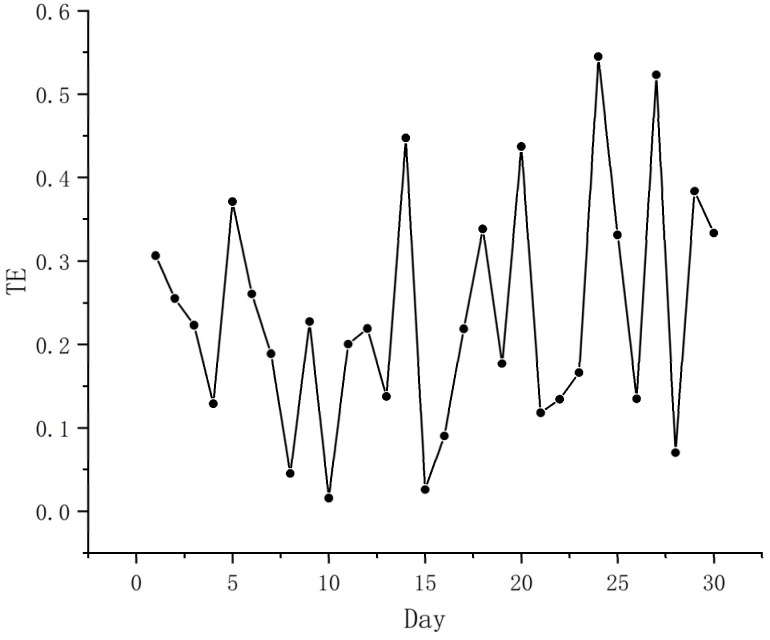
Causal relationship between positive and negative emotions.

**Figure 10 ijerph-19-12594-f010:**
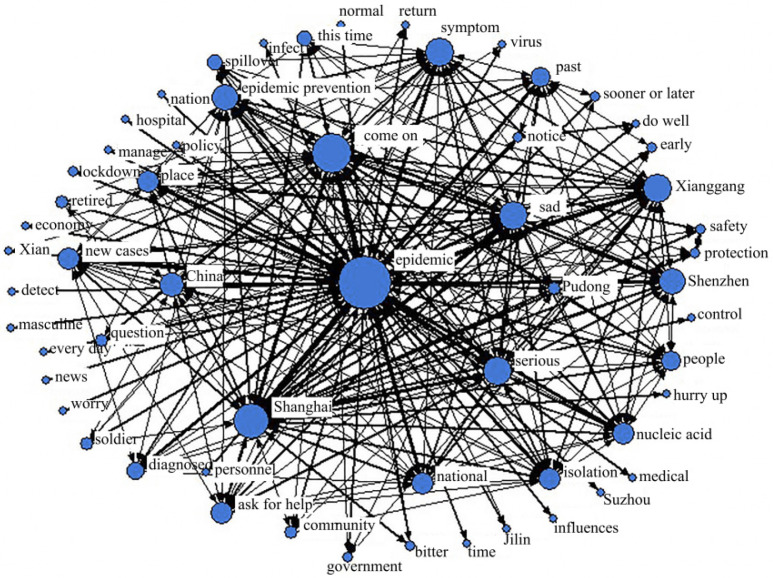
Social network analysis during emotional fermentation.

**Figure 11 ijerph-19-12594-f011:**
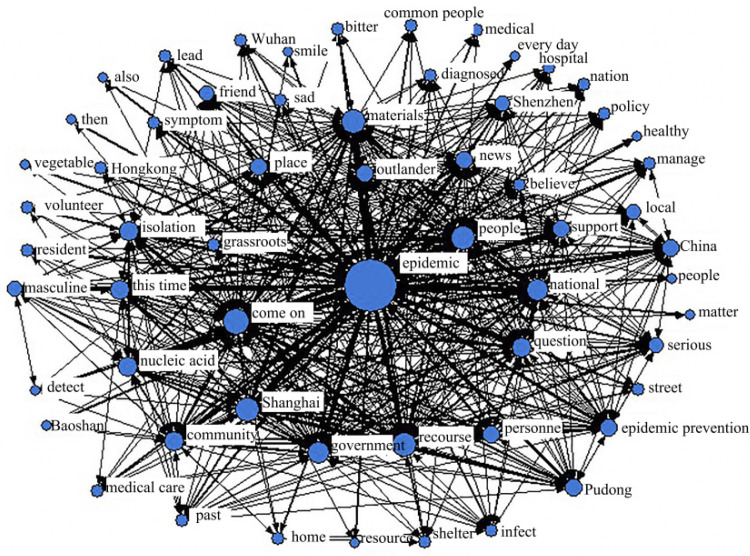
Social network analysis during the emotional peak period.

**Figure 12 ijerph-19-12594-f012:**
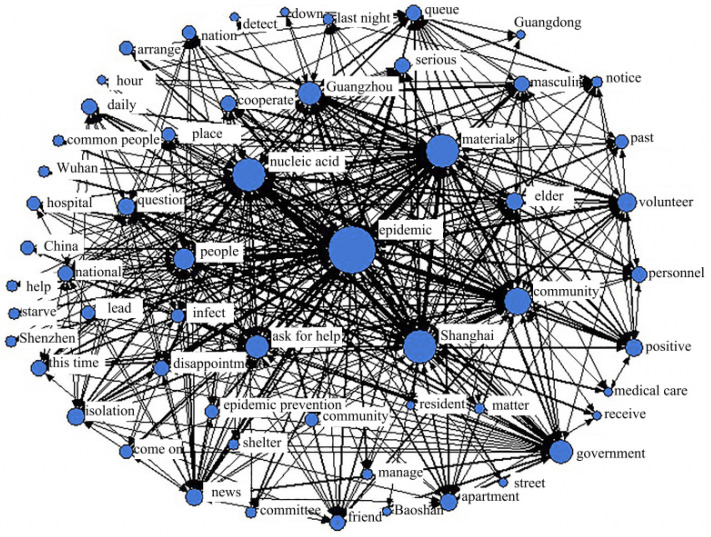
Social network analysis in the emotional chaos period.

**Table 1 ijerph-19-12594-t001:** Example of Weibo comment structure.

Comment Date	Reviewer Nickname	Title Link	Comment
10 March 2022	Goose egg sister is not softhearted	https://weibo.com/u/7338779976 (accessed on 20 April 2022)	Chengdu is almost free of the epidemic. We’re down to a low-risk area. Come on.
3 April 2022	Missing little kids	https://weibo.com/u/2036291735 (accessed on 20 April 2022)	Love your city and cooperate with all anti-epidemic arrangements. Fighting the epidemic together
5 April 2022	Cheese Pig Zyra	https://weibo.com/u/2608700454 (accessed on 20 April 2022)	I feel the management gap in the region. Baoshan has not distributed materials and parts of Pudong.

**Table 2 ijerph-19-12594-t002:** Word Frequency Weights in Weibo Comments.

Key Words	Frequency	tf	idf	tfidf	Key Words	Frequency	tf	idf	tfidf
Shanghai	19636	135.42	0.79	106.86	Thanks	1875	12.93	1.74	22.54
Epidemic	7781	53.66	1.15	61.97	Ask for help	1750	12.07	1.85	22.31
Come on	5027	34.67	1.36	47.30	Weibo	1711	11.80	1.79	21.17
Community	2989	20.61	1.58	32.60	Positive	1502	10.36	1.89	19.54
Nucleic acid	2941	20.28	1.60	32.44	Government	1530	10.55	1.81	19.14
Isolation	2829	19.51	1.62	31.58	Epidemic prevention	1455	10.03	1.87	18.79
Sad	2594	17.89	1.58	28.26	Shenzhen	1340	9.24	1.92	17.76
Anti-epidemic	2012	13.88	1.78	24.65	Safety	1380	9.52	1.84	17.50
Materials	2083	14.37	1.71	24.55	Bitter	1333	9.19	1.87	17.18
Shanghai residents	1930	13.31	1.76	23.49	Hospital	1259	8.68	1.96	17.03

**Table 3 ijerph-19-12594-t003:** Examples of sentiment classification content.

Emotion	Example
Gratitude	It’s hard work, the angels on the front line are hard work. Pay attention to protection and return safely. You guys are the best!
Confidence	I believe that the epidemic situation in Shanghai will soon see the light of day.
Sad	We haven’t started school in Shenzhen yet, sad! When the epidemic is over, it is estimated that another half semester will have passed. I am really heartbroken!
Anger	Are Shanghainese not Chinese? Half of the flight goes to Shanghai, have you ever thought that the life of Shanghai people is also life?
No emotion	The courier guys in Hangzhou should all be quarantined! Does it feel like the courier guys across the country have been quarantined?

**Table 4 ijerph-19-12594-t004:** Classification accuracy of each emotion.

Emotion Category	Accuracy	Recall	F1
Gratitude	0.9694	0.9596	0.9645
Confidence	0.9714	0.9533	0.9623
Sad	0.9184	0.9091	0.9137
Anger	0.7593	0.8454	0.8000
No emotion	0.8132	0.7551	0.7831

**Table 5 ijerph-19-12594-t005:** Number of various emotional comments.

Emotion Category		Quantity	Proportion	
Positive emotions	Gratitude	10,858	12.14%	19.81%
	Confidence	6864	7.67%	
Negative emotions	Sad	15,174	16.96%	42.64%
	Anger	22,975	25.68%	
No emotion		33,597	37.55%	37.55%

**Table 6 ijerph-19-12594-t006:** Daily changes in variables.

Date	Positive Emotions	Negative Emotions	Daily New Cases	Date	Positive Emotions	Negative Emotions	Daily New Cases
3.10	10	2	75	3.25	21	221	2269
3.11	210	219	83	3.26	190	226	2676
3.12	−85	−25	65	3.27	−191	379	3500
3.13	101	81	169	3.28	−75	−376	4477
3.14	340	240	139	3.29	−24	12	5982
3.15	−210	146	202	3.30	307	608	5653
3.16	198	−325	158	3.31	188	476	4502
3.17	−248	−166	260	4.1	−301	−603	6311
3.18	−40	19	374	4.2	373	1543	8226
3.19	−107	21	509	4.3	295	299	9006
3.20	383	137	758	4.4	482	−117	13,354
3.21	17	−109	896	4.5	−385	412	17,077
3.22	−149	24	981	4.6	−19	−161	19,982
3.23	−216	112	983	4.7	−79	−199	21,222
3.24	167	26	1609	4.8	20	527	23,624

## Data Availability

Not applicable.

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
