# Peer review of "Changes in Public Sentiment under the Background of Major Emergencies—Taking the Shanghai Epidemic as an Example"

_ijerph, 2022, doi:10.3390/ijerph191912594_

Round 1

Reviewer 1 Report

Dear authors,

This manuscript analyses the relationship between the COVID-19 pandemic and population sentiments. The study is conducted using the city of Shanghai as an example. Your manuscript is interesting but I need you to answer some questions:

INTRODUCTION

-          The introduction should not be to have subsections.

-          The authors have not stated the objective of the investigation.

METHODOLOGY

-          The authors must specify the research design. Simply saying "case study" is not valid in this case. This type of research is for specific cases with health problems that are noteworthy for some reason.

-          What was the target population? How was the sample chosen? The authors must specify it.

-          The authors must include the response rate of the participants in the study. All users were using Weibo? Were all accounts checked? How did you know that the accounts were not fake or had fraudulent use?

-          Have you consulted the ethics committee? The authors must mention and say the reference.

RESULTS

-          The heading of table 2 is not well understood. The authors should revise this.

DISCUSSION

-          It is not clear that the causal relationship is proven. To demonstrate causality, a longitudinal design must be used. The authors have not explained their design well and also say that they add new cases to analyse. It is not necessary to add new cases but to follow them up. The authors should justify their answers.

-          Do not include tables/figures in the "discussion". This is wrong. This is wrong. Authors should include figures in "results".

CONCLUSIONS

-          The authors should simplify the conclusions. They are very long.

Author Response

Thank you for your suggestion. My reply to you has been uploaded as an attachment.Please see the attachment.

Reviewer 2 Report

The paper “Changes in public sentiment under the background of major 2 emergencies—Taking the Shanghai epidemic as an example” tackles an interesting topic of shifts in public emotions, expressed in posts on social media, as a function of a pandemic. I find their methodology innovative and creative, potentially useful in future analyses of this sort, but in need of some refinement (details below). While the topic is indeed interesting, and the paper is fairly well written, I do have some reservations regarding the “input material” (emoticons and public posts) as well as the utterly explorative nature of the study (no theoretical framework whatsoever). Furthermore, I am not convinced that the findings of the study have any potential application in future pandemics.

Regarding the emoticons, the author stat that “….Expression symbols can well reflect the actual emotions of the commenter. For example, (symbol 1) represents sadness, (symbol 2) represents bitterness, and (symbol 3) represents gratitude. …”

This sounds reasonable enough – but are there any data regarding the perception of these emoticons? Are their meaning universally agreed upon? I am aware of the fact that the commenters on this platform are probably rather homogenous regarding the culture and education; nevertheless, the same emoticons can be interpreted quite differently by for example, different age groups (my students, for example, quite often comment on how their parents misinterpret the info they intended to convey when exchanging short messages accompanied by emoticons). It might also depend upon gender, user experience, etc. If possible, please cite the studies corroborating the notion that these symbols are indeed unambiguous.

Line 384, the authors state that: “When the number of daily new cases increased or decreased, positive and negative emotions increased or decreased accordingly” So, both positive and negative emotions shifted in the same direction, as a function of the number of new cases? Can you elaborate on that in more detail? What would be the psychological explanation for that? Is there any theoretical framework within which these findings can be explained?

It is hardly surprising that the impact of an epidemic is greater on the negative emotions than on the positive emotions, or that, as authors put it (line 392): “… during major adverse health events, the public is more likely to have negative emotions”

But what are the implications? Are there any suggestions regarding the implementation of some public policies or health campaigns? The authors did attempt to comment on this by stating (line 401 ) that “….the government or other personnel need to create more positive emotions for the public to suppress the growth of negative emotions”. But this is a rather vague suggestion, and it also brings to mind a related question: Is there any scenario in which governments wouldn't want to create more positive emotions? How is this specific to a pandemic?

In their further exploration of phases of the epidemic, authors continue to use ambiguous language and counterintuitive phrases like  “….the public still had a positive attitude towards the epidemic (line 430) public had a positive attitude towards the Shanghai epidemic. What does this mean in the context of this study? Positive attitudes towards epidemiological safety measures? Towards medical personnel? Or the government? Because, it is highly unlikely that people had positive attitude toward a contagious disease. The manuscript would benefit from use of a more precise language.

Author Response

(The authors gave the same response as above.)

Reviewer 3 Report

This study was interesting and it has contributions points for knowledge and filed. I have some comments that may help advancing the paper:

Introduction:

What is public sentiment, what is the definition?

How you define major emergencies and in your work how it relates to public sentiment? What are different perspectives about it?

There is no need to mention various instruments.

What is Weibo texts?

Was this study just considered gratefulness, confidence, sadness, anger, and no emotion? Explain. What does “no emotion” mean?

There is an explanation about emotion and learning but their relation to sentiment is vague.

Method:

Besides using the programs for analyzing the data, were there any people work on analyzing the comments and reach the teams or sentiment classification.

Results:

What is LAC? Needs more explanation and how does it works?

 Discussion:

It is mentioned that “In summary, it can be found that: first, changes in public sentiment under the epidemic were affected by government behavior”, where in the study the government behavior was addressed? What kinds of behaviors the authors mean?

Author Response

(The authors gave the same response as above.)

Round 2

Reviewer 1 Report

Dear authors,

The objective is confusing. You have made an introduction and then divided the objective into four specific objectives. You should set a standard objective and simplify everything you have written.

The chosen design is not valid. There is no such design. You should choose a standard qualitative or quantitative design.

The fact of tracking IP addresses I don't know if it is ethical or not. If the social network shares that data openly then nothing happens. However, it is not very common. Networks may know your location but it is not shared because it can cause personal security problems. This is controversial. You should justify this topic well.

The way to check if accounts are fake can be biased. The system you have used is not foolproof. It should be included in the "limitations of the study".

Author Response

Thank you for your suggestion. We modified as suggested and uploaded as an attachment.Please see the attachment.
